# Adapter-augmented Time Series Reconstruction for Source-free Domain Adaptation: A Black-box Method

## Abstract

Domain adaptation for time series classification is challenging due to the highly dynamic nature. This study addresses the most difficult subtask where both target labels and source data are inaccessible during adaptation, namely, source-free domain adaptation (SFDA). Although several promising approaches have been proposed, the problem remains under-explored. One issue is that most existing time series SFDA methods are tightly coupled with the architecture of the classification backbone, based on fine-tuning the backbone encoder to align with the source domain. In contrast, our method performs adaptation directly on the time series data rather than on latent features, treating the backbone classification network as a black box. This design significantly enhances the method's generality and applicability across different architectures. Specifically, we propose a coarse-to-fine adaptation framework: First, a source-pretrained reconstructor generates a base anchor that reflects domain-shared patterns. Second, a lightweight adapter is trained to further reduce the domain shift by jointly reducing the uncertainty of classification and the reconstructive error. Here, the adaptation is performed by updating only the adapter, while the full classification backbone remains frozen, allowing parameter-efficient fine-tuning based on learned priori from pre-training. Extensive experiments validate the state-of-the-art (SOTA) performance of the proposed method. Our codes are available at `https://anonymous.4open.science/r/ATSR-SFDA-52EB/`.

## 1 Introduction

Time series classification plays an important role in a wide range of applications (Wang et al., 2023; Gorbett et al., 2023; Mingyue et al., 2023), including human activity recognition (Xu et al., 2023; Hu et al., 2023; Kang et al., 2024; Ye et al., 2024), mechanical fault diagnosis (Tian et al., 2024; Luo et al., 2024; Qian et al., 2024), and EEG classification (Zhao et al., 2020; Pradeepkumar et al., 2024; Zhang et al., 2024). A major challenge to apply time series classification in real-world scenarios is context-related domain shift. For example, a pre-trained model for mechanical fault diagnosis will experience in general significant performance degradation when applied under different operating conditions, such as varying rotational speeds or torques compared to the pre-training conditions. This issue has attracted growing interest in unsupervised domain adaptation (UDA), which aims to transfer knowledge from a labeled source domain to an unlabeled target domain by aligning their distributions or learning domain-invariant representations for model reuse.

For UDA tasks, the source data remain accessible during adaptation. However, in many practical cases, accessing to the source data may be restricted due to storage limitations or privacy-sensitive concerns. This leads to the more challenging setting referred to as source-free domain adaptation (SFDA), where neither source data nor target labels are available during adaptation. In this work, we focus on SFDA for time series classification, which is a more challenging but less explored scenario in contrast to UDA.

In the literature, most SFDA methods for time series treat the pre-trained classification backbone as two components: An encoder followed by a classifier. During adaptation, the classifier is usually frozen, while the encoder is fine-tuned to map the target-domain data into a feature space in align-

ment with that of the source domain statistically, aiming to enforce the target fine-tuned encoder compatible with the source-pretrained classifier. Although this has become a mainstream paradigm for time series SFDA, such methods inherently assume full access to both the architecture and parameters of the pre-trained backbone, which might be impractical sometimes. Reliance on such white-box access not only tightly couples the adaptation process to the specific model structure but also gives rise to potential privacy risks, as sensitive source-domain information might be inferred or distilled from the accessible backbone. To address these limitations, we propose a concise but effective SFDA framework specifically designed for time series data, in which the entire pre-trained backbone is treated as a frozen black box during adaptation. As shown in Figure 1, our black-box approach allows the frozen model to take the time series arising from the reconstructive adaptation as input, without requiring access to the internal structure or parameters of the backbone.

Our method adopts a coarse-to-fine adaptation framework during the SFDA process. As shown in Figure 1, the first step performs a coarse-level adaptation, based on reusing the source-trained reconstructor on target data to matter domain-shared patterns. The resulting time series from this stage is referred to as base anchor as it anchors the source data of similar temporal patterns, which is a realistic priori as observed experimentally. The second step is the fine-level adaptation. Building upon the base anchor, we employ a dual-branch architecture to further reduce the source-target domain discrepancy. One branch, termed Source Inherited (SI) branch, directly inherits the base anchor, aiming to preserve the coarse adaptation result that reflects the domain-shared patterns. The other branch, called Target Compensation (TC) branch, takes the base anchor as input and uses an lightweight adapter to compensate for the divergence caused by the domain-private patterns. The finally adapted target sequence is obtained by combining the outputs from both branches.

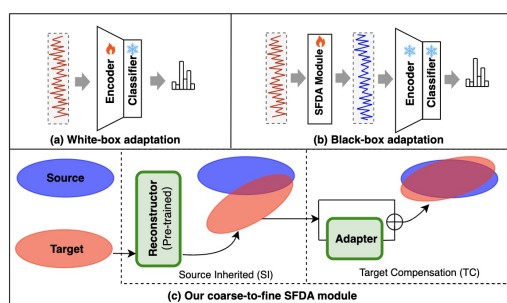

Figure 1: (a) Most SFDA methods apply white-box adaptation to modify the backbone encoder for source-target alignment in latent space; (b) Our method performs adaptation on the input data directly, treating the backbone classification network as a black box; (c) Our coarse-to-fine SFDA strategy for time series includes two phases termed source inherited on the basis of time series reconstruction and target compensation facilitated by adapter.

The contribution of this study is summarized as follows:

- Our method addresses source-free domain adaptation on time series at the data level rather than the representation level of the classification backbone, so it does not require access to or modification of the backbone's internal structure, making the approach universally applicable across different architectures as black boxes.

- We propose a coarse-to-fine adaptation framework. In the first step, the target domain data is transformed into a base anchor representing domain-shared patterns via a time series reconstructor. They are then refined through a target-specific compensation mechanism in the second step to achieve further alignment augmented by an adapter. Both the reconstructor and the adapter can have various implementations, making the framework universal.

- It is worth noting that the adaptation is merely based on fine-tuning the adapter on target data, while all the pre-trained modules including the time series reconstructor and the classification backbone are kept frozen. Adapter is known as a parameter-efficient fine-tuning technique, leveraging the learned knowledge from pre-training as a priori in augmenting time series reconstruction.

- Experimental results demonstrate the SOTA performance of our method on 3 widely used benchmarks.

## 2 RELATED WORKS

### 2.1 SOURCE-FREE DOMAIN ADAPTATION

In the unsupervised domain adaptation (UDA) setting He et al. (2023), both labeled source data and unlabeled target data are available during adaptation. Source-free domain adaptation (SFDA), in contrast, represents a more challenging and realistic scenario, where only unlabeled target data are accessible during the adaptation phase. Some existing approaches tackle SFDA by employing self-supervised fine-tuning strategies, and optimizing the decision boundary by minimizing entropy-based losses has been proven to be an effective strategy for domain adaptation Xia et al. (2022); Ahmed et al. (2021). Alternatively, some methods Du et al. (2021); Qiu et al. (2021) aim to generate synthetic data to approximate the source domain distribution, enabling the use of conventional UDA techniques Long et al. (2018); Wilson et al. (2020); Liu & Xue (2021) in a plug-and-play manner. Regarding domain alignment, a variety of matured methodologies have been widely explored, such as reconstruction-based approaches He et al. (2023), adversarial learning Wilson et al. (2023); Liu & Xue (2021), and contrastive learning-based strategies Meng et al. (2023), offering distinct advances in bridging the domain gap.

### 2.2 TIME SERIES SOURCE-FREE DOMAIN ADAPTATION

Although SFDA has been extensively studied in computer vision, its application to time series classification remains underdeveloped. MAPU Ragab et al. (2023b) addresses this issue by fine-tuning a source-pretrained encoder on a temporal imputation task, by which partially masked target time series is reconstructed to facilitate encoder-level alignment. TERSE Gong et al. (2025), specifically designed for multivariate time series SFDA, builds upon the MAPU framework by replacing the backbone network and introducing a graph-based spatial reconstruction task to capture and transfer the intrinsic spatial correlations among multivariate channels. TemSR Wang et al. (2024) presents a source-free approach that does not require additional source-domain pretraining. It leverages contrastive learning-based recovery to learn source-like feature distributions, based on which the encoder is trained for domain adaptation.

Despite their effectiveness, these time series SFDA methods rely on aligning encoder's representation with that of the source domain via unsupervised learning. As a result, they are tightly coupled with the architecture of the backbone network, limiting their generalizability and flexibility.

### 2.3 ADAPTER

Known as a parameter-efficient fine-tuning technique, adapter Houlsby et al. (2019) is originally proposed in natural language processing as a trainable plug-in middleware to transfer the internal representations of frozen Transformer downstream via supervised learning, and undergoes a series of advances by its variants Chen et al. (2022); Yin et al. (2023); Niu et al. (2023); Xin et al. (2024). It is the first time that adapter is applied as outside augmentation to refine time series reconstruction in an unsupervised manner, not in the form of a supervised plug-in ware.

## 3 METHODOLOGY

### 3.1 PRELIMINARIES

$\mathcal{D}_S = \left\{ X_S^i, y_S^i \right\}_{i=1}^{N_S}$ denotes a labeled dataset from source domain, where $X_S^i \in \mathbb{R}^{d \times L}$ represents uni-variate ($d = 1$) or multi-variate ($d > 1$) time series of length $L$, and $y_S^i \in \mathbb{R}^K$ the corresponding labels. $\mathcal{D}_T = \left\{ X_T^i \right\}_{i=1}^{N_T}$ denotes an unlabeled dataset from target domain, which shares the same label space $y = \{1, 2, \cdots, K\}$ with $\mathcal{D}_S$. We follow the common settings of SFDA to assume that the marginal distributions $P_S(X_S) \neq P_T(X_T)$ due to feature shift, and it is strictly prohibited to access source data during adaptation.

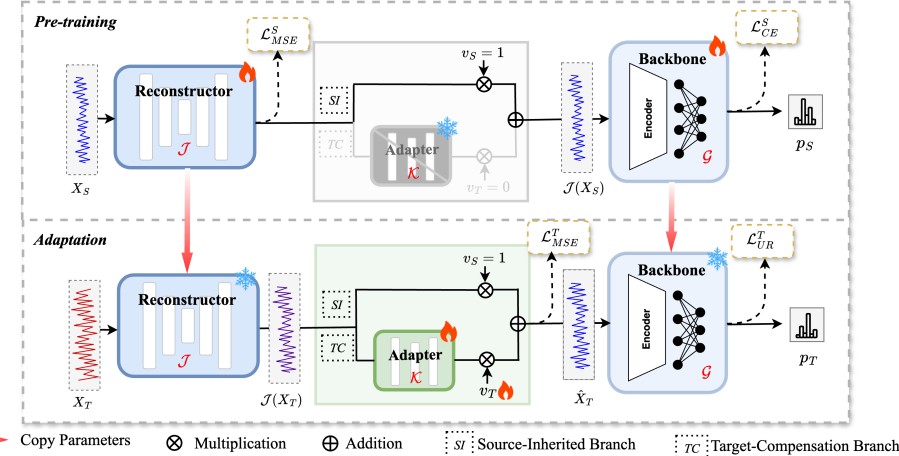

Figure 2: Network architecture of our coarse-to-fine black-box adaptation method. In the pre-training phase, the time series reconstructor is trained on source data and the outcome is applied to the backbone network as input to train it into a reusable classification model, where the optimization objective is the reconstruction error and the classification loss, respectively. In the adaptation phase, the reconstructor is directly inherited to perform time series reconstruction on target data, and augmented by fine-tuning an adapter in addition to the inheritance, for which the optimization objective is to reduce the uncertainty loss of classification while control the reconstruction error to certain extent. Here, our full adaptation is merely based on fine-tuning the adapter on target data, and no additional operation is needed, offering a parameter-efficient solution.

## 3.2 OVERALL ARCHITECTURE

The goal of SFDA is to maximize classification precision on the target domain, or minimize the target error $\epsilon_T$ . According to the classical error bound theory for domain adaptation Ben-David et al. (2010), the target error $\epsilon_T$ has an upper bound:

$$\epsilon_T \leq \epsilon_S + \frac{1}{2}d_{\mathcal{H}\Delta\mathcal{H}}(\mathcal{D}_S, \mathcal{D}_T) + \lambda \tag{1}$$

where $\epsilon_S$ denotes the source domain error, and $\lambda$ the minimum error achievable by the ideal model. As $\lambda$ is fixed, to reduce $\epsilon_T$, we should minimize both $\epsilon_S$ and the domain discrepancy $d_{\mathcal{H}\Delta\mathcal{H}}$ to the best extent. We approach this in a framework of coarse-to-fine adaptation.

As the pipeline shown in Figure 2, reduction of $\epsilon_S$ is achieved in the pre-training phase. In our design, the components requiring pre-training prior to adaptation include not only the backbone network for time series classification, but also an encoder-decoder-based reconstructor to produce the input to the backbone. Both are trained on source data and remain frozen thereafter. As regular temporal patterns allow finer time series reconstruction than random ones, smaller reconstruction error means better promise for classification. Therefore, we apply time series reconstruction as a part of the adaptation, which is also used in He et al. (2023) for UDA task, but relying on fine-tuning on target data, not frozen to undergo adapter-based refinement as ours. $\epsilon_S$ can then be reduced by optimizing classification over reconstruction.

In the adaptation phase, we firstly apply the pre-trained time series reconstructor on the target data to produce an anchor bearing domain-sharing patterns as the base for data transferring, and then design a dual-branch network structure to reduce the domain discrepancy $d_{\mathcal{H}\Delta\mathcal{H}}(\mathcal{D}_S, \mathcal{D}_T)$ caused by domain-private patterns, where the two parallel branches termed SI branch and TC branch will jointly yield the finally synthetic output for classification. The SI branch directly inherits the ancestral adaptation resulting from the pre-trained reconstructor. The TC branch pipelines two trainable components: An autoencoder-based adapter and a learnable scaling factor to control the strength of compensation over SI. The TC branch is the unique module to be trained during adaptation while all the others are kept frozen, making the adaptation parameter-efficient.

### 3.3 PRE-TRAINING ON SOURCE DOMAIN

As shown in Figure 2, we pre-train jointly a classification backbone and an accompanying time series reconstructor based on source data, where the reconstructed time series serves as the input to the classification backbone. The reconstructor has a encoder-decoder structure, learning to replay the temporal patterns of the source domain data. The reconstruction loss to be optimized is as follows: $\mathcal{L}_{MSE}^{S} = \frac{1}{N_S} \sum_{i=1}^{N_S} \left\| X_S^i - \mathcal{J}(X_S^i) \right\|^2$, where $\mathcal{J}$ represents the reconstructor, frozen once pre-trained.

As shown in MAPU Ragab et al. (2023b), one-dimensional convolutional neural network (1D-CNN) performs well in time series classification tasks. For the sake of comparison, we use the same network structure as applied in MAPU to train the classification backbone. We apply the output from the pre-trained reconstructor as the input to the downstream classification task and minimize the cross entropy (CE) loss in training.

### 3.4 COARSE-TO-FINE ADAPTATION

As illustrated in Figure 2, during adaptation, we first obtain a coarse-level adaptation as anchor, corresponding with source-target shared patterns. Based on this anchor, we further refine the adaptation through a dual-branch subsequent transferring, yielding the finally synthetic sequence fed to the frozen classification backbone for inference.

#### 3.4.1 SOURCE INHERITED BRANCH.

Due to domain overlap, the target and source domains are not entirely incompatible. During pre-training, the reconstructor has learned the temporal patterns of the source data. We observe that reconstruction of the original target time series using this reconstructor, namely $\mathcal{J}\left(X_T^i\right)$, has already aligned with the source domain to some extent due to the existence of domain-common temporal patterns. Therefore, the coarse adaptation can serve as a source-anchored reliable base for fine adaptation.

In fine adaptation, we preserve the base anchor $\mathcal{J}\left(X_T^i\right)$ as partial composition of the finally synthetic sequence. Specifically, we employ a residual connection to ensure that the anchor contributes directly to the final reconstruction output.

#### 3.4.2 TARGET COMPENSATION BRANCH.

The base anchor alone is insufficient to make the pre-trained backbone to produce reliable classification results, due to the existence of domain-private patterns other than domain-common patterns, which causes source-target divergence if applying the reconstructor directly. So, we need fine-level adaptation to further reduce such divergence.

Given the base anchor $\mathcal{J}\left(X_T^i\right)$ as ancestral adaptation, we further introduce a lightweight module $\mathcal{K}$, say adapter, to provide target-specific compensation for reducing the discrepancy between the base anchor and the true source domain.

Additionally, we apply two scaling factors, $v_S$ and $v_T$, to SI branch and TC branch, respectively, to balance the composition of both branches at the final end of reconstruction. During pre-training, $v_T$ and $v_S$ are fixed at 0 and 1, respectively. In the adaptation phase, $v_T$ is trainable and $v_S$ is still fixed at 1 (only adjustable in test time adaptation). The final output incorporating the two branches can be formulated as:

$$\hat{X}_T^i = v_T \cdot \mathcal{K}\left(\mathcal{J}\left(X_T^i\right)\right) + v_S \cdot \mathcal{J}\left(X_T^i\right) \tag{2}$$

#### 3.4.3 LOSS FUNCTION.

In fine-level adaptation, our task is to make the reconstructed time series possess regular temporal patterns rather than random ones by reducing the reconstruction error, while improve the confidence of classification by imposing loss-directed constraint on the reconstruction. Tsallis entropy is a useful measure for evaluating the confidence of classifying unlabeled target time series, making it effective for reducing classification uncertainty in unsupervised settings. The improved version of Tsallis entropy has been shown to be suitable for minimizing the uncertainty reduction loss in SFDA

tasks Xia et al. (2022). Optimizing this loss encourages the adaptation model to generate synthetic target sequences that lead to more confident and consistent predictions using the pre-trained classification backbone. The original Tsallis entropy loss is defined as:

$$\mathcal{L}_{Tsallis} = -\frac{1}{q-1}\frac{1}{K}\frac{1}{N_T}\sum_{i=1}^{N_T}[\sum_{k=1}^{K}\delta(\mathcal{G}^k(\hat{X}_T^i))]^q \tag{3}$$

where $\delta(\mathcal{G}^k(\hat{X}_T^i))$ is the $k$th softmax output of $\mathcal{G}(\hat{X}_T^i)$ and $q$ is the power of the probability, usually set to 2 in such task.

The improved Tsallis entropy Xia et al. (2022) enhances its generalization capability and has been shown to be more suitable for SFDA setting. The uncertainty reduction loss based on the improved Tsallis entropy is formulated as:

$$\mathcal{L}_{UR}^T = -\frac{1}{q-1}\frac{1}{K}\sum_{i=1}^{N_T}\sum_{k=1}^{K}\frac{\gamma_i(\eta_i^k)^q}{\beta_k} \tag{4}$$

where

$$\eta_i^k = \frac{\exp(\mathcal{G}^k(\hat{X}_T^i)/\tau)}{\sum_{j=1}^{K}\exp(\mathcal{G}^j(\hat{X}_T^i)/\tau)} \quad\text{(5a)} \qquad \gamma_i = \frac{N_T[1+\exp(-E(\eta_i))]}{\sum_{j=1}^{N_T}[1+\exp(-E(\eta_j))]} \quad\text{(5b)}$$

$$\beta_k = \sum_{i=1}^{N_T}\eta_i^k \tag{6}$$

Eq.5a represents temperature rescaling, where the temperature parameter $\tau$ is set to 2. Eq.5b introduces a confidence-based sample weighting scheme, where the weights are dynamically assigned according to the entropy of each sample. Specifically, $E(\eta_i) = -\sum_{k=1}^{K}\eta_i log\eta_i$ is the entropy of the prediction $\eta_i$, where lower entropy corresponds to higher weights. Eq.6 aims to mitigate the impact of class imbalance, normalizing class contributions to ensure fair learning across all categories. These modifications makes it more robust in reflecting the confidence of classification.

Since the uncertainty reduction loss aims at optimizing the decision boundary, it may lead to suboptimal or trivial solutions Vu et al. (2019). To ensure meaningful adaptation, we use reconstruction loss to augment it. This loss serves to regularize the randomness of the temporal patterns in the input fed to the classification backbone to prevent the uncertainty reduction loss from producing confident but trivial solutions. The reconstruction loss is defined as:

$$\mathcal{L}_{MSE}^T = \frac{1}{N_T}\sum_{i=1}^{N_T}\left\|X_T^i - \hat{X}_T^i\right\|^2 \tag{7}$$

Then, the overall optimization objective becomes:

$$\mathcal{L}_{overall}^T = \phi\mathcal{L}_{UR}^T + \mathcal{L}_{MSE}^T \tag{8}$$

where $\phi$ is a hyperparameter.

## 4 EXPERIMENT

### 4.1 DATASETS AND EXPERIMENTAL SETUP

We evaluate our method using 3 time series datasets organized in ADATIME Ragab et al. (2023a), which are collected from real-world application scenarios, including machine fault diagnosis (MFD), sleep stage classification (SSC), and human activity recognition (UCIHAR). The 3 datasets, as summarized in Appendix, are heterogeneous in terms of time length and number of channels. Our solution utilizes U-net and autoencoder (AE) as the reconstructor and adapter, respectively, to turn out all the reported performances if not otherwise declared. Note that the framework is universal, not exclusively implemented as such.

Table 1: Comparison of macro F1 scores (%) on MFD, SSC, and UCIHAR benchmarks.

| Algorithm | | MFD Benchmark | | | | | | SSC Benchmark | | | | | | UCIHAR Benchmark | | | | | |
|---|---|---|---|---|---|---|---|---|---|---|---|---|---|---|---|---|---|---|---|
| | SF | 0→1 | 1→0 | 1→2 | 2→3 | 3→1 | AVG | 16→1 | 9→14 | 12→5 | 7→18 | 0→11 | AVG | 2→11 | 12→16 | 9→18 | 6→23 | 7→13 | AVG |
| DDC | ✗ | 74.50 | 48.91 | 89.34 | 96.34 | 100.0 | 81.82 | 55.47 | 63.57 | 55.43 | 67.46 | 54.17 | 59.22 | 60.00 | 66.77 | 61.41 | 88.55 | 77.29 | 75.67 |
| DCoral | ✗ | 79.03 | 40.83 | 82.71 | 98.01 | 97.73 | 79.66 | 55.50 | 55.35 | 55.35 | 67.49 | 53.76 | 59.12 | 67.20 | 64.58 | 54.38 | 89.66 | 90.46 | 84.10 |
| HoMM | ✗ | 80.80 | 42.31 | 84.28 | 98.61 | 96.28 | 80.46 | 55.51 | 63.49 | 55.46 | 67.50 | 53.37 | 59.06 | 83.54 | 63.45 | 71.25 | 94.97 | 91.41 | 84.10 |
| MMDA | ✗ | 82.44 | 49.35 | **94.07** | **100.0** | **100.0** | 85.17 | 62.92 | 71.04 | **65.11** | 70.95 | 43.23 | 62.79 | 72.91 | 74.64 | 62.62 | 91.14 | 90.61 | 81.40 |
| DANN | ✗ | 83.44 | 51.52 | 84.19 | 99.95 | **100.0** | 83.82 | 58.68 | 64.29 | 64.65 | 69.54 | 44.13 | 60.26 | 98.09 | 62.08 | 70.70 | 85.60 | 93.33 | 84.97 |
| CDAN | ✗ | 84.97 | 52.39 | 85.96 | 99.70 | **100.0** | 84.60 | 59.65 | 64.18 | 64.43 | 67.61 | 39.38 | 59.04 | 98.09 | 61.20 | 71.30 | 96.73 | 93.33 | 86.79 |
| CoDATS | ✗ | 67.42 | 49.92 | 89.05 | 99.21 | 99.92 | 81.10 | 63.84 | 63.51 | 52.54 | 66.06 | 46.28 | 58.44 | 86.65 | 61.03 | 80.51 | 92.08 | 92.61 | 85.47 |
| AdvSKM | ✗ | 76.64 | 43.81 | 83.10 | 98.85 | **100.0** | 80.48 | 57.83 | 64.76 | 55.73 | 67.58 | **55.19** | 60.21 | 65.74 | 60.52 | 53.25 | 79.63 | 88.89 | 74.67 |
| SHOT | ✓ | 41.99 | 57.00 | 80.70 | 99.48 | 99.95 | 75.82 | 59.07 | 69.93 | 62.11 | 69.74 | 50.78 | 62.33 | **100.0** | 70.76 | 70.19 | **98.91** | 93.01 | 86.57 |
| NRC | ✓ | 73.99 | 74.88 | 69.23 | 78.04 | 71.48 | 73.52 | 52.09 | 58.52 | 59.87 | 66.18 | 47.55 | 56.84 | 97.02 | 72.18 | 63.10 | 96.41 | 89.13 | 83.57 |
| AaD | ✓ | 71.72 | 74.33 | 78.31 | 90.07 | 87.45 | 80.58 | 57.04 | 65.27 | 61.84 | 67.35 | 44.04 | 59.11 | 98.51 | 66.15 | 68.33 | 98.07 | 89.41 | 84.09 |
| MAPU | ✓ | **99.43** | 77.42 | 85.78 | 99.67 | 99.97 | 92.45 | **63.85** | **74.73** | 64.08 | **74.21** | 43.36 | 64.05 | **100.0** | 67.96 | 82.77 | 97.82 | **99.29** | 89.57 |
| **ATSR(ours)** | ✓ | 99.34 | **89.55** | 88.17 | 98.6 | 100.0 | **95.13** | 63.5 | 72.07 | 61.87 | 71.62 | 52.36 | **64.28** | 100.0 | **86.88** | **87.45** | 94.29 | 91.04 | **91.93** |

We follow the setting in MAPU Ragab et al. (2023b) to use the same classification backbone and the given domain transfer scenarios on the MFD, SSC, and UCIHAR data, allowing our method to be made comparable to MAPU and its baselines. The evaluation metric is also kept consistent with MAPU, using the macro F1 score. We use Adam optimizer with a batch size of 32. To balance the magnitudes of the two loss terms, we set $\phi$ to 0.1 for all the datasets. Other details are available in Appendix.

We built our model based on Ubuntu 20.04.5 with Python 3.9 and NVIDIA GeForce RTX 2080Ti (11GB) GPU. The total number of the parameters of our main solution is less than 34.74 million, with 34.53 million from the reconstructor, 8,170 from the adapter, and 0.2 million from the classification backbone network. The model is computationally efficient to enable 10,293.44 million floating-point operations per second (FLOPS). Since only adapter based fine-tuning is required during adaptation, its parameter-efficient nature makes the deployment easy.

## 4.2 PERFORMANCE EVALUATION

Adopting the baselines from MAPU, we compare our method with the conventional UDA methods, including DDC Tzeng et al. (2014), DCoral Sun et al. (2017), HoMM Chen et al. (2020), MMDA Rahman et al. (2020), DANN Ganin et al. (2016), CDAN Long et al. (2018), CoDATS Wilson et al. (2020), and AdvSKM Liu & Xue (2021), as well as the 4 SFDA methods, including SHOT Liang et al. (2020), NRC Yang et al. (2021), AaD Yang et al. (2022), and MAPU Ragab et al. (2023b). The results shown in Table 1 includes the macro F1 scores of the 5 domain transfer scenarios, and the average over all, where SFDA against UDA is marked as "√" and "✗", respectively, and ATSR (Adapter-augmented Time Series Reconstruction) refers to our method.

Our method outperforms all baselines on the 3 datasets, leading to the SOTA performance of 95.13%, 64.28%, and 91.93% in terms of macro F1 score, which means 2.89%, 0.35%, and 2.63% improvement on the 3 benchmarks, respectively. The improvements are obvious, especially when the previous SOTA performances achieved by MAPU reach a high standard at 92.45% and 89.57% on the MFD and UCIHAR benchmarks. Yet, no method works so fine on the SSC benchmark, including our time series reconstruction based adaptation, because the random nature of such EEG data makes precise temporal pattern modeling difficult.

Table 2: Advantage of the dual-branch design.

| Macro F1 score(%) | MFD | SSC | UCIHAR |
|---|---|---|---|
| w/o SI branch | 73.26 | 50.97 | 48.25 |
| w/o TC branch | 77.84 | 59.10 | 86.00 |
| Full model | **95.13** | **64.28** | **91.93** |

Table 3: Comparison of coarse-to-fine adaptation to fine-tuning reconstructor only.

| Macro F1 score(%) | MFD | SSC | UCIHAR |
|---|---|---|---|
| Fine-tune only | 91.37 | 63.04 | 86.47 |
| Coarse-to-fine | **95.13** | **64.28** | **91.93** |

### 4.3 ABLATION STUDY

#### 4.3.1 ARCHITECTURE AND IMPLEMENTATION.

To validate the rationality of our dual-branch design, we provide comparative results in Table 2. In the absence of the SI branch, the model fails to preserve the domain-shared patterns, making the sole adapter unable to achieve effective domain adaptation. Meanwhile, without the TC branch, the coarse-level adaptation alone is insufficient to approach satisfactory performance.

In addition, we compare the proposed coarse-to-fine adaptation to fine-tuning directly the reconstructor for full adaptation. The results summarized in Table 3 show that the coarse-to-fine adaptation achieves superior performance. Moreover, since our approach only updates the lightweight adapter and the scaling factor during adaptation, it is much more efficient than fine-tuning the reconstructor.

Table 4: Comparison of component-level network implementation from coarse to fine.

| Macro F1 score(%) | MFD | SSC | UCIHAR |
|---|---|---|---|
| AE + AE | 89.56 | 57.63 | 26.65 |
| AE + U-net | 91.03 | 56.75 | 29.19 |
| U-net + U-net | 91.90 | 60.94 | 87.12 |
| U-net + LSTM | 93.06 | 63.23 | 89.10 |
| LSTM + AE | 94.09 | 63.75 | 90.51 |
| U-net + AE | **95.13** | **64.28** | **91.93** |

We also compare different configurations in realizing reconstructor and adapter. As shown in Table 4, the first column lists different implementations of reconstructor and adapter in order. The results demonstrate that the proposed framework is universally promising with versatile implementations but U-net and AE realized reconstructor and adapter are most competitive.

#### 4.3.2 LOSS SELECTION.

As demonstrated in Table 5, when only $\mathcal{L}_{MSE}^{T}$ is used as the optimization objective, the performance drops notably. Meanwhile, when adaptation is guided solely by $\mathcal{L}_{UR}^{T}$, the performance also degrades.

Table 5: Comparison of loss functions' impact on adaptation.

| Macro F1 score(%) | MFD | SSC | UCIHAR |
|---|---|---|---|
| $\mathcal{L}_{UR}^{T}$ only | 93.36 | 60.24 | 88.50 |
| $\mathcal{L}_{MSE}^{T}$ only | 86.19 | 57.83 | 84.66 |
| $\mathcal{L}_{UR}^{T}$ & $\mathcal{L}_{MSE}^{T}$ | **95.13** | **64.28** | **91.93** |

Table 6: MMD variation of our method.

| Dataset | MFD | SSC | UCIHAR |
|---|---|---|---|
| Before adaptation | 0.1624 | 0.1004 | 0.1217 |
| Adapted | **0.0702** | **0.0417** | **0.0818** |

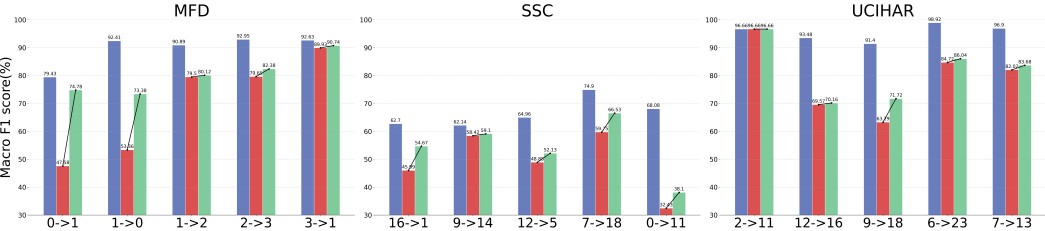

Figure 3: Macro F1 scores when replacing the 1D-CNN backbone with TCN. Blue, red, and green bars represent performance on source domain, and that on target domain before and after adaptation, respectively.

#### 4.3.3 BLACK-BOX NATURE.

In our method, the backbone network is treated as a black box. To verify the generality of our approach across different backbone architectures, we conduct experiments by replacing the original

1D-CNN backbone with a Temporal Convolutional Network (TCN), while keeping all the other settings consistent with those previously described. As illustrated in Figure 3, our method is promising if comparing before to after adaptation, demonstrating its effectiveness across different backbone architectures.

### 4.3.4 FEATURE VISUALIZATION.

We visualize the features yielded by the backbone encoder. Specifically, to illustrate the role of the SI branch, we compare the features of the target data before and after passing through the source-pretrained reconstructor, against the source features. As shown in Figure 5, the SI branch significantly reduces the distance between the features of the target domain samples and their counterparts in source domain. This evidences the rationality of using the reconstructed target data as the base anchor in aligning the source-target common features.

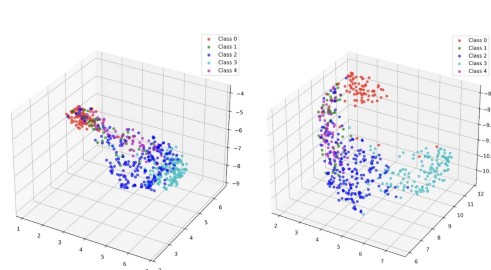

Figure 4: Feature distributions before (left) and after (right) applying adapter-based compensation on SSC data.

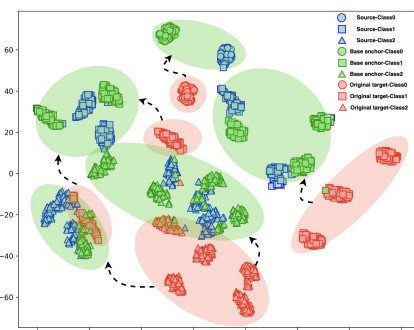

Figure 5: Visualization of the transfer learning effect of Source Inherited branch. We use t-SNE to visualize the features yielded by the backbone encoder in a given scenario (MFD 0→1). Data from source domain, target domain, and SI branch are shown in blue, red, and green, respectively.

As shown in Figure 4, we compare the dimensionality-reduced features before and after applying the TC branch to the base anchor. We observe that TC results in better-separated clusters to allow a clearer decision boundary.

### 4.3.5 STATISTICAL EVIDENCE.

Following the theoretical insights of Ben-David et al. (2010), the target error in domain adaptation is closely related to $d_{\mathcal{H}\Delta\mathcal{H}}(\mathcal{D}_S, \mathcal{D}_T)$, say, the discrepancy between source and target domains. A practical approach is to minimize the Maximum Mean Discrepancy (MMD) Gretton et al. (2006) between source and target feature distributions. In our full model, although MMD is not directly optimized, we find that our approach implicitly reduces the MMD between the two domains, as shown in Table 6. This fact manifests why our approach can lower the upper bound of the target error $\epsilon_T$ as defined in Eq.1.

## 5 CONCLUSION

This study focuses on source-free domain adaptation (SFDA) for time series classification. By treating the pre-trained model as a black box, we propose a coarse-to-fine adaptation framework that operates directly on the time series data. Extensive experiments demonstrate the rationality and universality of our design as well as its SOTA performance.

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

# A APPENDIX

## A.1 TEST TIME ADAPTATION (TTA)

During the adaptation phase, the scaling factor $v_S$ on the SI branch is fixed at 1. Considering that different target samples exhibit varying degrees of discrepancy with respect to the source domain, we propose to perturb the SI ratio at test time to check its impact on classification stability and alter its value on the fly to gain better robustness.

Let $\Delta$ represent a small perturbation on $v_S$, and the step-wise perturbation becomes

$$v_S^j = 1 \pm j \cdot \Delta, j = 0, 1, 2, \cdots, n \tag{9}$$

where $n$ refers to the maximum span for parameter variation. Let $p_j^i$ represent the output of classifying $X_T^i$ conditional on the scaling factor $v_S^j$ and $s_j^i = CosSim(p_{j-1}^i, p_j^i)$ the cosine similarity to measure the classification stability when changing $v_S^{j-1}$ to $v_S^j$. After scanning the whole span of $[1 - n \cdot \Delta, 1 + n \cdot \Delta]$, we apply softmax function to obtain the corresponding weights $w_j^i$ as:

$$w_j^i = \frac{e^{s_j^i}}{\sum_{k=-n}^n e^{s_k^i}} \tag{10}$$

The classification is then finalized as

$$p^i = \sum_{j=-n+1}^{n} w_j^i p_j^i \tag{11}$$

According to Table 7, we find that TTA can lead to a slight performance improvement. Here, we also try replacing the cosine similarity with entropy. Yet, the TTA method using cosine similarity achieves the highest improvement, with performance gains of 1.0%, 0.55%, and 0.24% on the 3 datasets, respectively. The step-wise perturbation is set to 0.1% and the maximum searching span is $\pm 10$ steps.

Table 7: Impact of TTA based on cosine similarity or entropy against no TTA.

| Macro F1 scores(%) | MFD | SSC | UCIHAR |
|---|---|---|---|
| w/o TTA | 95.13 | 64.28 | 91.93 |
| TTA by Entropy | 96.10 | **64.83** | 91.93 |
| TTA by CosSim | **96.15** | **64.83** | **92.17** |

## A.2 ABOUT LARGE LANGUAGE MODELS

The translation, writing, and polishing of the paper were assisted by using an LLM (such as Chat-GPT).

