# OpenReview forum: "Adapter-augmented Time Series Reconstruction for Source-free Domain Adaptation: A Black-box Method"
_ICLR.cc/2026/Conference — Submitted to ICLR 2026_

### Official Review · Reviewer_LsgH · 2025-10-27

**Soundness:** 3
**Presentation:** 3
**Contribution:** 2
**Rating:** 2
**Confidence:** 3

**Summary:**

This work presents a method for source-free domain adaptation of time series data. This is a recently-introduced setting for domain adaptation in which the source data is not available at adaptation-time, meaning that the method cannot leverage traditional approaches such as distributional alignment to adapt new domains of data. The authors also claim that the method works for black-box time series models. Their method uses an adapter at test time to alter the time series data to be more fitting for the classifier originally trained on source data. There is significant discussion throughout the paper on the parameter efficiency of this adapter approach. They show experiments on several time series datasets, demonstrating near-SOTA performance on the tasks for these datasets.

**Strengths:**

- This work directly tackles a novel problem for time series domain adaptation, the case in which source data are not available at adaptation time. This is particularly important for models that might be trained on proprietary data or foundation models trained on large datasets where source data cannot easily be accessed.
- The method and models used are well-described. The architecture of their domain adaptation method is described clearly and in-full, allowing the reader to fully understand the approach.
- The authors properly motivate the intuition behind their method, especially the reasoning for the adapter and residual connections to modify the target domain data directly on the time series space.
- The authors conduct several experiments on datasets from diverse domains, and their ablations target the key components of the architecture. Their method achieves SOTA performance across the datasets in which they tested their method. The paper could serve from some more datasets to establish absolute supremacy though, but the experiments in the paper seem to establish at least some weak signal that the method is better than baselines in this setting.
- It is encouraging to see that the ablations show that the method overall performs the best, especially given that this method combines somewhat-disparate components together.

**Weaknesses:**

- This work fundamentally lacks novel advancements over previous approaches. Adapter tuning has been well-studied in many fields, and other methods, particularly in UDA, have proposed to change the structure of the time series data to perform the domain adaptation. The novelty of this method relies on the adapter approach applied in this setting, which is an incremental step. This could be justified by large improvements in performance over baselines, but only marginal increases are shown in the table. The authors need more argument as to the novelty of the method.
- As a conceptual problem, while the authors claim that the method is “black-box”, this is not entirely true. While it is true that the model could be trained with any internal architecture, the model must be trained with an encoder-decoder style network with a classifier on the output of the decoder. This limits the applicability of this method to black-box classifiers unless they follow these specifications. The meaning of “black-box” should be better defined in the paper, as an argument can be made around the extent of “black-box” that this approach can handle.
- While the authors make significant claims around the efficiency of their adaptation method, the computational efficiency of the proposed architecture is still quite higher than the efficiency of training the classifier alone. As the authors state on page 7, 34.53m/34.74m are in the reconstructor alone; very few of the parameters are in the classifier. This must be taken into account when considering the computational efficiency of the method overall, and I don’t believe that this method can necessarily be considered “efficient”.
- No error bars are shown in Table 1, making it difficult to assess if the model's boost in performance is statistically-significant.

**Questions:**

- This problem setting is fairly narrow and specific to cases in which the data are not available during adaptation. Could the authors clarify on real-world scenarios when this setting is absolutely necessary?
- It is not entirely clear why the reconstruction is necessary. Is it possible that the authors could train the adapter on the source data instead of the reconstructed data to then feed into the classifier? I believe experiments are necessary to establish the need for this component.
- The authors mention that $v_T$ is trainable at adaptation time. What does the model end up learning for this value? This seems to be an important component as it controls the extent to which the “adapted” part of the source domain is fed in. It is good that this value could potentially change based on the accuracy of the classifier for the severity of the domain shift, but empirical results would be needed to prove this.
- Related to the last question, authors need to show the performance of no domain adaptation to the domain transfer settings in Table 1. What would be the performance if trained on the first domain and transferred to the second with no test-time adaptation? This can be shown only for overlapping target variables since the UDA conditions still hold.

---

### Official Review · Reviewer_YyMW · 2025-10-31

**Soundness:** 3
**Presentation:** 3
**Contribution:** 2
**Rating:** 2
**Confidence:** 3

**Summary:**

While it is encouraging to address the SFDA problem in a black-box manner, the proposed method lacks sufficient technical novelty and contribution to warrant acceptance.

**Strengths:**

1.	This paper presents a compact yet powerful SFDA framework tailored for time series data, where the pre-trained backbone remains fully frozen and is treated as a black box during the adaptation process.

**Weaknesses:**

1.	The proposed method trains an encoder–decoder-based reconstructor using source data, which may appear to violate the SFDA setup.

2.	The experiments are conducted on only three datasets (MFD, SSC, and UCIHAR), which seems insufficient to validate the effectiveness of the proposed method.

3.	The domain adaptation performance improvement achieved by the proposed method is not significant.

**Questions:**

1.	The one-dimensional convolutional neural network (1D-CNN) employed in the proposed method is adopted from many existing works (i.e., MAPU), suggesting that no novel model architecture is introduced by the authors.

2.	Apart from incorporating a reconstruction loss into the existing Tsallis entropy for SFDA, it is unclear what technically novel components are contributed by the proposed method.

3.	In Table 1, only selected domain adaptation cases (e.g., 0 → 1) are reported, while others cases are omitted. Without presenting results for all adaptation cases, the experimental results cannot be considered reliable.

---

### Official Review · Reviewer_ofq2 · 2025-10-31

**Soundness:** 2
**Presentation:** 3
**Contribution:** 2
**Rating:** 4
**Confidence:** 4

**Summary:**

This paper proposes an interesting method that performs adaptation directly on the time series data rather than on latent features, which boost the method's generality.

**Strengths:**

Strengths:
1. This paper proposes a black-box method for adaptation at the data level in the time series SFDA for the first time, avoiding dependence on the internal structure and parameters of the classification backbone network. This solves the problem of poor generality in existing methods and represents an important breakthrough in the field of SFDA.
2. The paper achieves domain adaptation by fine-tuning lightweight Adapters while freezing all pre-trained modules. This not only significantly reduces computational costs but also fully utilizes the prior knowledge of pre-trained source domains, demonstrating excellent engineering design.
3. The paper successfully combines theory with practice, capturing coarse-grained domain shared patterns by generating "basic anchors" with pre-trained reconstructors, and then handling fine-grained domain private patterns through "target compensation" with Adapters, which is logically clear and in line with the domain adaptation problem.

**Weaknesses:**

1. The core strength of the paper lies in its black-box universality, however, the paper lacks in-depth theoretical analysis of this universality. As an PEFT technique, the effectiveness of Adapter usually depends on the internal structure of the pre-trained model, such as Transformer layers. This paper applies Adapter to data reconstruction, which is a good usage, but its theoretical advantages compared to traditional feature space Adapter and the universality for different backbone networks like CNN-based and Transformer-based should be more strictly argued. The authors should supplement analysis, adding an in-depth discussion of the theoretical advantages of black-box adaptation and data space alignment in the methodology section, explaining how Adapter formalizes the reduction of divergence through enhancing time series reconstruction, rather than merely empirically minimizing uncertainty loss.
2. The reconstructor is pretrained on the source domain, aiming to capture the temporal patterns of the source domain. Applying it directly to generate "basic anchors" for target domain data depends strongly on the similarity of temporal patterns between the source and target domains. If domain shift mainly manifests as structural changes in temporal patterns rather than simple statistical shifts, the "anchors" generated by the reconstructor may carry strong source domain bias. The authors should supplement experimental analysis, exploring whether the quality of the basic anchors and the compensating ability of Adapter will decrease in tasks with significant differences in time series structure, such as frequency, periodicity, and location of mutation points.
3. The SFDA process in the paper, where this module introduces additional reconstructors and Adapters, as well as a dual-branch structure. Particularly in the pretraining stage, it is necessary to train a reconstructor and a classification backbone network. This part requires experiments to verify a detailed comparison of the total training time, i.e., pretraining + adaptation time and the inference time during the adaptation stage. Compared to mainstream SFDA methods, how is the actual computational efficiency of this black-box method? Especially on multivariate time series, will the computational complexity of the reconstructor become a bottleneck?
4. The issue of hyperparameter sensitivity in the loss function, where the loss function during the adaptation phase is a weighted sum of reconstruction loss and uncertainty loss, including weight hyperparameters, this type of weighted loss is usually very sensitive to hyperparameters. The impact of their values on the final classification accuracy needs to be elaborated, as well as whether these hyperparameters can be adaptively adjusted based on certain statistical characteristics of the target domain data, in order to enhance the robustness of the method.

**Questions:**

As in Weaknesses

---

### Official Review · Reviewer_sCC1 · 2025-11-01

**Soundness:** 3
**Presentation:** 3
**Contribution:** 2
**Rating:** 4
**Confidence:** 3

**Summary:**

The paper proposes ATSR, a source-free domain adaptation method for time series classification that treats the pre-trained backbone as a black box. The approach uses a coarse-to-fine framework: first applying a pre-trained reconstructor to generate a base anchor, then using a lightweight adapter for target-specific compensation. The method achieves SOTA results on three benchmarks.

**Strengths:**

1. Novel black-box perspective: The paper addresses a limitation of existing SFDA methods that require white-box access to backbone architectures. This is practically important for privacy and deployment scenarios.
2. Well-motivated design: The coarse-to-fine framework is intuitive - using reconstruction to capture domain-shared patterns then adapting for domain-specific differences makes conceptual sense.
3. Parameter efficiency: Only fine-tuning the adapter while keeping other components frozen is computationally efficient and follows modern fine-tuning best practices.
4. Strong experimental results: Achieves SOTA performance across datasets with notable improvements.

**Weaknesses:**

1. Superficial theoretical justification. How does reconstruction specifically reduce dH∆H? the authors fail to rigorously establish why data-level reconstruction should minimize the H-divergence term more effectively than feature-level adaptation.
2. The paper's novelty appears to be primarily architectural rather than methodological. Time series for domain adaptation exists, adapters are well-established, and the improved Tsallis entropy loss is borrowed from Xia et al. (2022). The authors' contribution seems limited to combining these existing components in the source free scenario.

**Questions:**

1. Some equations use different font styles for same variables. For example, J for the reconstructor. Inconsistent subscript/superscript usage for indices
2. How sensitive is the method to the quality of the pre-trained reconstructor? What happens if the source-trained reconstructor is poorly trained?'
3. Why does the method perform poorly on SSC dataset

---

### Meta-Review · Area_Chair_mxqd · 2026-01-05

**Summary:**

The reviewers generally agree that the paper addresses a relevant problem (source-free domain adaptation for time series) and is clearly written. However, the main concerns consistently raised include limited technical novelty, insufficient theoretical justification, overstated black-box claims, and incomplete experimental validation. The method largely combines existing components (reconstruction, adapters, entropy-based objectives) without delivering a clear methodological or theoretical advancement. These concerns collectively informed the recommendation to reject.

**Reviewer Concerns:**

The authors did not submit a rebuttal, and therefore none of the reviewers’ substantive concerns were addressed.

Key outstanding issues include:

Lack of clear technical novelty beyond combining existing techniques.

Insufficient theoretical analysis explaining why data-level reconstruction effectively reduces domain shift.

Ambiguity and limitations in the claimed “black-box” setting, due to architectural and pretraining assumptions.

Limited and incomplete experimental evaluation, including missing adaptation cases and lack of statistical analysis.

Questionable efficiency claims given the additional cost of the reconstructor.

**Reviewer Scores:**

Given the absence of a rebuttal and no clarification or additional evidence provided, it is unlikely that reviewers would have increased their scores.

---

### Decision · Program_Chairs · 2026-01-26

Reject